# Advanced Adrenocortical Carcinoma: From Symptoms Control to Palliative Care

**DOI:** 10.3390/cancers14235901

**Published:** 2022-11-29

**Authors:** Elena Ruggiero, Irene Tizianel, Mario Caccese, Giuseppe Lombardi, Ardi Pambuku, Vittorina Zagonel, Carla Scaroni, Fabio Formaglio, Filippo Ceccato

**Affiliations:** 1Pain Therapy and Palliative Care with Hospice Unit, Veneto Institute of Oncology IOV-IRCCS, 35128 Padua, Italy; 2Department of Medicine DIMED, University of Padova, 35128 Padova, Italy; 3Endocrine Disease Unit, University-Hospital of Padova, 35128 Padova, Italy; 4Department of Oncology, Oncology Unit 1, Veneto Institute of Oncology IOV-IRCCS, 35128 Padua, Italy

**Keywords:** adrenocortical carcinoma, symptom management, palliative care, early palliative care, multidisciplinary evaluation

## Abstract

**Simple Summary:**

Adrenocortical cancer (ACC) is a rare malignancy, often diagnosed late and with a poor prognosis. Currently, ACC best management is achieved in referral centers, where a multidisciplinary approach (endocrinologists, oncologists, surgeons, radiologists and radiotherapists) can intercept the course of a patient with ACC early and operate with life-prolonging intents. Even in cases of advanced disease, multimodal treatments (chemotherapy and mitotane, surgery and/or radiotherapy) and skillful management of the medical complications of ACC can ensure significant improvements in survival. However, patients with advanced ACC suffer from relevant psychophysical symptoms and experience significant losses in quality of life. There is now robust evidence that the early integration of supportive and palliative care in standard oncological management may relieve cancer patients’ burden, mediate aggressive treatments and improve quality of life, and not only in the end-of-life period. In this paper, we provide an up-to-date literature review on the role of supportive and palliative care in ACC management.

**Abstract:**

The prognosis of patients with advanced adrenocortical carcinoma (ACC) is often poor: in the case of metastatic disease, five-year survival is reduced. Advanced disease is not a non-curable disease and, in referral centers, the multidisciplinary approach is the standard of care: if a shared decision regarding several treatments is available, including the correct timing for the performance of each one, overall survival is increased. However, many patients with advanced ACC experience severe psychological and physical symptoms secondary to the disease and the cancer treatments. These symptoms, combined with existential issues, debase the quality of the remaining life. Recent strong evidence from cancer research supports the early integration of palliative care principles and skills into the advanced cancer patient’s trajectory, even when asymptomatic. A patient with ACC risks quickly suffering from symptoms/effects alongside the disease; therefore, early palliative care, in some cases concurrent with oncological treatment (simultaneous care), is suggested. The aims of this paper are to review current, advanced ACC approaches, highlight appropriate forms of ACC symptom management and suggest when and how palliative care can be incorporated into the ACC standard of care.

## 1. Introduction

Adrenocortical carcinoma (ACC) is a rare malignancy, with a reported incidence of 0.7–1.3 cases per million/year [1,2]. The prognosis is variable from its presentation. Positive prognostic markers are stage, pathological grading and expertise of the center [3,4]. In clinical practice, ACC is often discovered as a metastatic disease with a poor prognosis: the five-year survival is below 28% in most series [5]. Common clinical features of ACC include steroid secretion and mass-related symptoms, often combined. The first characterizes the endocrine aspects of patients: cortisol or androgen excess varies from mild to overt secretion, including severe Cushing’s syndrome and female virilization [5,6].

According to the European Society of Endocrinology (ESE) and European Network for the Study of Adrenal Tumors (ENSAT) guidelines, a multidisciplinary team evaluation is recommended [5]. A multidisciplinary team, with the primary focus on life-prolonging management, is the standard of care for patients with ACC: it includes periodical meetings involving, at the least, endocrinologists, medical and radiation oncologists, surgeons, pathologists, radiologists and genetic counsellors [3,5]. Several systemic antineoplastic regimens, extended resection and local treatments may be used in patients with advanced ACC, ultimately impairing the wellbeing and the quality-of-life (QoL) of patients with cancer and their families–caregivers [6,7,8].

Supportive and simultaneous approaches involve the prevention and management of the adverse events/effects of cancer and cancer-related treatment in patients with good survival prognoses in short- to mid-term time periods. Palliative care is the sum of interventions intended to provide relief from cancer-related symptoms and stress in order to improve QoL (for patients with worsening clinical condition) and relieve cancer patients’ burden [9]. There is now robust evidence that the standard life-prolonging approach to patients with cancer should overlap with the early integration of supportive, simultaneous and palliative care.

European ESE/ENSAT [5] and American Association of Clinical Endocrinology (AACE) [3] guidelines recommend the early integration of palliative/simultaneous care in the multidisciplinary evaluation of ACC patients. In referral centers for endocrine and oncological diseases, a palliative team should be present and a close collaboration (which, in most cases, will not yet have been started) between the ACC multidisciplinary group and the palliative team is necessary. To the best of our knowledge, although the integration of standard ACC treatment with supportive and palliative care is of utmost importance, there are no data regarding their application in patients with ACC.

This narrative review focuses on the description of simultaneous care and its integration in the standard management of patients with ACC.

## 2. Endocrine Treatment: Mitotane and Steroidogenesis Inhibitors

Mitotane is the approved drug for the treatment of ACC and it is used both in the adjuvant setting and with advanced/metastatic ACC [10,11]. In patients with advanced disease, combined therapy with etoposide, doxorubicin, cisplatin (EDP) and mitotane (EDP-M) represents the current first-line treatment. The evidence is limited because there are few studies (most of which are observational) reporting the association between mitotane and chemotherapy in the treatment of advanced ACC [5,12]. Guidelines from the ESE/ENSAT and AACE suggest mitotane monotherapy for ACC patients without a residual tumor following surgery who have a perceived high risk of recurrence after radical surgery (as in the case where Ki67 >10%) and association of mitotane with chemotherapy (EDP-M) in advanced cases with poor prognosis [13]

Mitotane is a lipophilic drug derived from dichlorodiphenyltrichloroethane with a strong steroidogenesis inhibition effect: steroidogenic acute regulatory protein (StAR) and CYP11A1 are the targets of mitotane action, partially explaining its strong adrenolytic effect [14]. The commercially available mitotane formulation is 500 mg tablets (Lysodren^®^ in Europe). Mitotane treatment is started as soon as possible after surgery [5] with a low-dose (3 g/day after 12 days) or high-dose (3 g/day after 2 days, 6 g/day after 12 days) regimen, achieving similar plasma levels and demonstrating similar onset of adverse events [15]. Some patients do not achieve the therapeutic plasma concentration of 14–20 mg/L in the follow-up, even at doses up to 6–7 g/day (12–14 tablets in divided doses), due to poor water solubility, the large volume of distribution, the onset of adverse effects and inter/intra-individual variability [16]. Moreover, it has been found that 40% of unchanged mitotane could be detected in the feces 12 h after an oral intake of a single 2 g mitotane dose in tablet form [17]. The most common mitotane-induced side effects in patients with ACC are gastrointestinal disturbances, neurologic symptoms, leukopenia and hepatic disorder (liver failure is rare; however, asymptomatic increases in hepatic enzymes—in particular, gamma-GT—are common) [5], alone or combined: they represent a major limit to treatment adherence [15,16,17]. Novel oral (recrystallization from microemulsion, nanosuspension, liposomal) and injectable (micellar) formulations are currently being developed in order to improve the efficacy and tolerability of mitotane [16].

Substitutive glucocorticoid replacement is recommended for all patients during mitotane treatment, with the exception of cases that still present glucocorticoid excess. Due to the strong induction effect on CYP3A4 activity and the increase in cortisol-binding-globulin (CBG), ACC patients treated with mitotane need higher substitutive doses for effective replacement [11,15].

It is recommended that treatment with mitotane should begin as soon as possible after surgery (ideally within 6 weeks) with an escalating regime depending on the patient’s performance status, aiming to reach a plasma concentration between 14 and 20 mg/L (these values are proposed in several consensuses and guidelines; nonetheless, some experts suggest an enlarged range of 8–30 mg/L). Mitotane plasma levels should be periodically assessed. After the first assessments (more or less monthly at the beginning of treatment), mitotane dosage should be adjusted according to plasma concentrations and tolerability and considering that the threshold for side effects can be different from the suggested therapeutic range (levels >20 mg/L may be well-tolerated and patients can experience side effects with 14 mg/L). In patients without recurrence who tolerate mitotane therapy, it is recommended to administer the drug for at least 2 years [18] and up to 5 years [5,19].

Steroidogenesis inhibitors are adjuvant in the treatment of hypercortisolism, which is the most frequent presentation of hormonal excess in ACC patients (almost 50–60%) [20,21]. Guidelines recommend that every patient with suspected ACC should be evaluated for hormonal hypersecretion with clinical and biochemical assessments at baseline and during follow-up. Hypercortisolism in ACC patients can be severe and appear as rapidly developing Cushing’s syndrome with resistant arterial hypertension, hypokalemia, new-onset diabetes, mood disorders, immunosuppression, bone frailty and wasting symptoms. In a minority of cases, ACC can be associated with androgen excess (virilization in women) [19]. Overt hypercortisolism is detrimental and leads to increased morbidity and mortality due to its complications: opportunistic infections, cardiovascular events and wasting syndrome. In the case of severe hypercortisolism, mitotane therapy alone is not enough to control hormonal levels because it takes several weeks to reach efficacy. Therefore, adjuvant therapies, such as steroidogenesis inhibitors (ketoconazole, metyrapone and osilodrostat) or a glucocorticoid receptor antagonist (mifepristone), are suggested [22].

Despite the lack of studies, metyrapone is considered the first choice for advanced ACC with severe Cushing’s syndrome due to the rapid onset of its effects [23]. It is well-tolerated, it can be administered in association with mitotane and EAP and its metabolism is not affected by mitotane [23]. Metyrapone is an oral formulation and the dosage employed for the treatment of Cushing’s syndrome varies between 500 and 6000 mg/day depending on the severity of hypercortisolism [24]. Due to its short half-life (up to 4 h), it requires multiple administrations daily. It acts by inhibiting CYP11B1 and CYP11B2, which has the consequence of reducing aldosterone and cortisol secretion [25]. Due to the accumulation of precursors with mineralocorticoid activity, metyrapone treatment can be associated with worsening hypertension and hypokalemia; in the long term, it leads to hirsutism and/or acne, so it is less preferable in women with androgen-secreting ACC [26].

Ketoconazole is a steroidogenesis inhibitor approved for the treatment of Cushing’s syndrome that, due to its short half-life (3–4 h), is orally administered in fractionated daily doses of 200 to 1200 mg/day [26]. Its major adverse effect is hepatotoxicity, demonstrated by an elevation in liver enzymes that occurs early at the start of treatment or at dose up-titration. It is recommended not to start treatment if hepatic liver enzyme levels are > threefold greater than normal and to discontinue treatment if they increase. However, in clinical practice, increased liver enzymes should not preclude ketoconazole prescription [27]. Due to its inhibiting effects on several enzymes involved in adrenal steroidogenesis, it leads to hypogonadism, and its long-term use is less preferred in men. Ketoconazole has a strong inhibiting effect on CYP3A4, which leads to multiple drug interactions [25].

It is important to point out that, once mitotane plasma levels have reached the therapeutic window, the dose can be reduced due to its pharmacokinetics (the reported plasma elimination half-life is 18–159 days [16]) and fat reservoir (it is a lipophilic drug): it is recommended to use the lowest and best-tolerable mitotane dose that is able to guarantee a level of >14 mg/L in the long-term follow up [5]. Moreover, if hypercortisolism is well-controlled, the dosages of the steroidogenesis inhibitors can, by relying on clinical status and biochemical measurements, also be reduced (until their withdrawal, since mitotane is also a steroidogenesis inhibitor and acts as a cortisol-lowering drug below the therapeutic range) in order to avoid adrenal insufficiency and reduce the number of pills required per day. A decade ago, Kamenicky et al. assessed the feasibility of concomitant treatment with mitotane, ketoconazole and metyrapone in patients with Cushing’s syndrome and severe acute complications. They found out that this combination therapy was a valid alternative to surgical bilateral adrenalectomy in critical patients [28]. Moreover, this therapeutic approach can also be useful in optimizing other types of palliative care for ACC patients with advanced disease, severe symptoms of hypercortisolism and the need for rapid control of the disease.

In some cases, severe hypercortisolism in patients with ACC is a life-threatening condition that represents a management challenge [29]. Osilodrostat, a recently developed inhibitor of adrenal 11 beta-hydroxylase, is effective in the treatment of severe Cushing’s syndrome resulting from ACC due to its rapid onset of action, safety and limited drug interactions. The starting dose depends on the severity of the hypercortisolism and whether there is a need to initiate antineoplastic treatment; in the latter case, the “block and replace” strategy is effective, given the high risk of osilodrostat-induced adrenal insufficiency [30].

Mifepristone, originally developed as a progesterone receptor antagonist, is currently used as a glucocorticoid receptor antagonist, with its effects prompting a rise in circulating cortisol levels [22]. Mifepristone provides a rapidly effective and valuable option for patients with severe hypercortisolism when surgery is unsuccessful or impossible: it can achieve swift reductions in body weight, blood pressure, glucose metabolism and most Cushingoid appearances [31]. However, it requires close monitoring of potentially severe hypokalemia and of the clinical signs of adrenal insufficiency [32].

## 3. Standard Life-Prolonging Curative Treatments for Patients with ACC

Patients with ACC often require combined multimodal treatment, personalized and shared after a multidisciplinary discussion (see Figure 1). In patients undergoing surgical resection and with localized disease, the only approved systemic adjuvant treatment is mitotane (discussed in Section 2). There is currently no solid evidence to support other systemic treatments in the adjuvant setting for patients with ACC. Considering the high recurrence rate, it might be reasonable to evaluate adjuvant chemotherapy treatment in patients with a high risk of relapse, neoplastic thrombosis or locally infiltrating disease. Currently, the level of evidence present in the literature does not allow definitive conclusions.

Two phase III randomized trials (ADIUVO-2 and ACACIA; NCT03583710 and NCT03723941, respectively) in the active and recruitment phases, respectively, are ongoing, aiming to evaluate the efficacy of platinum-based chemotherapy treatment with or without mitotane in patients with high-risk resected ACC stage I–III and Ki67 ≥ 10%. Obviously, no data are yet available. In the locally advanced or metastatic disease setting, however, there are several studies that have evaluated different chemotherapy schemes, with single drugs or combinations, and found encouraging results [5,33]. A small multicenter study enrolled 28 patients treated with EDP-M and obtained promising results (ORR 53.5%; 95% CI 35–72) [34]. This small study paved the way for the randomized phase III FIRM-ACT trial [35], which cleared the use of the same triplet of chemotherapy drugs plus mitotane in patients with locally advanced or metastatic ACC. Treatment with EDP-M showed a higher response rate (23.2% vs. 9.2%, *p* < 0.001) and a higher median progression-free survival (mPFS) (5.0 months vs. 2.1 months; HR 0.55; 95% CI 0.43–0.69; *p* < 0.001) than the control arm with streptozotocin plus mitotane (SZ-M). In terms of overall survival (OS), no statistically significant differences were found between the two treatment arms (14.8 months and 12.0 months, respectively; hazard ratio, 0.79; 95% CI 0.61–1.02; *p* = 0.07). These results could be justified by the possibility of cross-over that was offered to patients with progressive disease after the first line of treatment. EDP-M is presently considered the standard of care for the treatment of patients with locally advanced or metastatic ACC [5,12], and supportive treatments are essential to mitigate EDP-M side effects [36].

For patients progressing after EDP-M treatment, the oncological therapeutic alternatives currently remain very limited. Several studies have evaluated various drugs/schemes, such as streptozotocin, gemcitabine in combination with capecitabine and temozolomide, and obtained modest results [37,38,39]. The need for better comprehension of the molecular pathways involved in the genesis and progression of ACC has led to the study of different target therapies. Unfortunately, most of the studies that evaluated tyrosine kinase inhibitors (sunitinib, erlotinib, gefitinib and sorafenib) were small studies that did not report particularly encouraging response rates [33]. A randomized phase III study evaluated linsitinib, an IGF-1 receptor inhibitor, and enrolled 139 patients with locally advanced or metastatic ACC who progressed after standard chemotherapy treatment. Linsitinib did not improve OS compared to the placebo [40]. Cabozantinib, a multi-kinase inhibitor (VEGFR2, MET and RET), was evaluated in a small study with 16 patients previously treated for recurrent ACC (after mitotane discontinuation). At the end of the study, three patients had a partial response and five patients had a stable disease [41]. Two phase II studies (NCT03370718 and NCT03612232) evaluating the use of cabozantinib in patients with ACC are ongoing.

The immunotherapy approach—in particular, with the use of immune checkpoint inhibitors—has been evaluated in patients with advanced ACC. Pembrolizumab, an anti-PD1 immune checkpoint inhibitor, was evaluated in two phase II studies, resulting in disease control rates of 52% and 56%, respectively. In both studies, the efficacy of this therapeutic approach was independent of PD-L1 expression, microsatellite instability/mismatch-repair deficiency and tumor-infiltrating lymphocytes (TILs) [42,43]. Excessive cortisol secretion is able to counteract the immune system and reduce immunotherapy efficacy: we previously reported an impressive response to anti-PD-1 immunotherapy in a young male patient with advanced ACC with mismatch-repair deficiency and adrenal insufficiency [44]. The anti-PD1 monoclonal antibody nivolumab was also studied in a small multicenter phase II study that enrolled ten patients: in this study, only modest activity with an mPFS of <2 months was demonstrated [45]. The greater availability of extended genomic analysis techniques (next-generation sequencing) could pave the way for novel personalized treatment, as effective therapeutic alternatives are currently very limited. Some studies with the aim of personalizing treatments and improving the outcomes for these patients have already been published [46,47] and others are in progress.

In clinical practice, all options for life-prolonging treatment (surgery, mitotane, chemotherapy and radiotherapy) should be considered. The goal is remission for stage I–III ACC; however, a cure is not possible for metastasized patients. In these patients with advanced ACC, life-prolonging treatment attacking the tumor burden should be provided, in accordance with the performance status, with a shared decision involving the patients and their caregivers. Therefore, a careful balance of the patient’s performance status (combining mass-related symptoms and endocrine aspects) and the impact of treatments (especially chemotherapy or extended resection during open surgery) should be tailored to match the appropriate patient: a multidisciplinary evaluation can define whether a patient is fit for the therapy and supportive treatment.

## 4. Symptom Management

Most oncological patients experience several physical and psychological disorders in the course of their diseases, which are often clustered, simultaneous and occur in such a way as to fuel each other [48]. Highly prevalent symptoms of advanced cancer patients include anorexia, fatigue, anxiety, depression, nausea, vomiting, constipation, pain, confusion and breathlessness [49]. The symptom burden is directly related to patients’ functionality and to the perceived QoL for both patients and their families. Moreover, increased disease and symptom burdens negatively impact the appropriate chemotherapy dose, leading to reduced survival [50]. Therefore, intensive symptom management is mandatory in cancer patients. Cancer management guidelines recommend the assessment of pain and other symptoms in oncological practice [51,52], making it possible to improve symptom monitoring over time and allowing early identification of patient needs [53].

### 4.1. Fatigue and Adrenal Insufficiency

Fatigue is an unpleasant subjective symptom that incorporates total body feelings ranging from tiredness to exhaustion, creating an unrelenting overall condition that interferes with an individual’s ability to function at normal capacity [54]. Fatigue is among the most common and distressing symptoms in patients with cancer and usually one of the more difficult to manage since it becomes worse in the advanced phase of the disease [55]. The prevalence of fatigue in ACC patients is high in clinical practice, reflecting the overall activity of the hypothalamic–pituitary–adrenal axis (fatigue is a common effect of hypercortisolism and adrenal insufficiency), and it is perceived as a troublesome symptom.

Fatigue is the expression of several combined pathological mechanisms provoked directly by cancer and antineoplastic treatments: malnourishment; anemia; vitamin and micro-elemental depletion; electrolyte disorders (especially calcium, magnesium, potassium and sodium); diabetes; hypothyroidism (which can be also mitotane-induced in patients with ACC [56]); renal, liver, cardiac or pulmonary insufficiency; hypoxemia; depression; spiritual crisis; and physical activity restrictions [55,57,58]. However, many of the mechanisms underlying fatigue are yet to be unveiled. Central roles for immune system misbalance, dysregulation of the hypothalamus–pituitary–adrenal axis, alteration of circadian rhythms, loss of muscle mass, and abnormalities in the ATP cycle have been hypothesized [59]. There is evidence in the literature that variations in inflammation-related genes may be risk factors for fatigue, suggesting a genetic contribution [55]. Fatigue during ACC treatments, particularly mitotane and/or chemotherapy, is the main factor limiting compliance and cannot be explained entirely by adrenal insufficiency.

The possibility of achieving effective control over fatigue is low, even after the correction of the underlining conditions. Corticosteroids are currently the most commonly used drugs; nonetheless, the synthetic steroid type and dose must be carefully selected for patients with ACC (usually, high-dose substitutive glucocorticoid therapy is required for mitotane-induced adrenal insufficiency). Non-pharmacological treatments may include physical exercise, acupuncture and meditation [60].

### 4.2. Nausea and Vomiting

Nausea and vomiting are common symptoms in cancer patients and can occur alone or simultaneously. They are also among the most common side effects of chemotherapy and radiotherapy, and not only in advanced ACC patients.

According to data published by the European Medicine Agency, nausea, vomiting, diarrhea and anorexia are very common adverse effects during mitotane treatment (≥1/10 patients) [5]; the prevalence of nausea has been found to increase to 90% after EDP-M [36]. Mitotane, a lipophilic drug with poor oral bioavailability of 30–40% [16], is better absorbed following the intake of high-fat nutrients (milk, chocolate or yogurt) [5]. In clinical practice, nausea and vomiting (which can evolve into mucositis [17]) can appear early with a low plasma concentration of 5 mg/L [16]. The combination of the number of medications required, the large size of the tablets, altered taste, nausea and reduced appetite ultimately limit the achievement of the therapeutic range.

Some of the possible consequences of nausea and vomiting are metabolic alterations and electrolyte imbalance, non-adherence to therapy, malnutrition and cachexia. Chemotherapeutic agents and radiotherapy protocols are classified according to their ability to induce nausea and vomiting. Moreover, nausea is one of the most frequent side effects of analgesic opioid therapy, imposing the need for a careful balance of beneficial effects and limiting side effects [50].

Anti-dopaminergic drugs can be used for patients with nausea during ACC treatment: metoclopramide (especially for mitotane-induced nausea), domperidone, haloperidol, levosulpiride and other neuroleptics can effectively control nausea and vomiting. If a single agent is ineffective, switching to another treatment is warranted. Refractory nausea can be treated by adding second-line treatments, such as corticosteroids, benzodiazepines or antiserotoninergic drugs with behavioral adaptations. In radiotherapy- and chemotherapy-induced nausea, ondansetron and other antiserotoninergic drugs and neurokinin receptor antagonists (i.e., aprepitant, fosaprepitant), alone or in combination with antidopaminergics, have shown high efficacy. For refractory vomiting from radio- and chemotherapy, cannabis extracts can be considered [48]. Loperamide is suggested for mitotane-induced diarrhea.

### 4.3. Anorexia and Cachexia

Anorexia and cachexia are multifactorial syndromes that depend on individual factors, cancer type, disease stage and treatment protocol. They are the result of the cancer microenvironment and chronic inflammation [61,62,63]. In patients with advanced cortisol-secreting ACC, hormonal dysregulation and metabolic abnormalities, such as insulin resistance, increased proteolytic activity and lipolysis, can worsen anorexia and cachexia [64]. In clinical practice, changes in appetite, or in the taste of food with subsequent loss of appetite, are commonly reported after starting mitotane (and poorly described in the literature). Moreover, cortisol-induced visceral adiposity [65] correlates with weight loss and muscle mass loss, key factors for decreased treatment response and survival in cancer patients [65,66,67]. Malnourishment and sarcopenia are strongly related to reduced tolerance of chemotherapy, increased risk of postoperative complications and deterioration of QoL. Thus, malnutrition and cachexia strongly affect survival: clinical data suggest that almost 20% of deaths are attributable to malnutrition rather than cancer progression [63,68]. Nutritional support for cancer patients is a major aspect of palliative care: minimal goals could be body weight maintenance and prevention of further weight loss [69]. However, at the end-of-life period, nutritive supplements and artificial nutrition do not impact positively on either survival or QoL [70].

Hormonal orexigenic drugs are used to improve appetite and reduce weight loss in advanced cancer patients, ultimately resulting in improvements in QoL (as reported in a recent meta-analysis [71]). Their prothrombotic effect is the major limiting risk in their use. Anecdotally, steroids (for brief treatments), cyproheptadine and cannabinoids can be used to increase appetite. Cannabis extracts are licensed in Italy, with some restrictions, for the treatment of refractory cancer anorexia [68,72]. Oral ghrelin mimetic and other orexigenic drugs that stimulate appetite are in the development pipeline [73,74].

### 4.4. Depression and Neurological Side Effects

Most patients with cancer have a depressed mood, and this disorder gets worse when approaching the end-of-life period. Some are not able to cope with the existential issues that arise from shorter survival and do not develop sufficient adaptive behavior to overcome the devastating impact of cancer on their lives. Finally, many patients develop recognizable psychiatric diseases; notably, depression [75]. Advanced ACC patients hold many additional organic risk factors that increase the risk of the onset of depression; in particular, metabolic and endocrine alterations, especially in cortisol-secreting ACC [76], and the results of oncological treatments [75]. Moreover, chronic pain and disability resulting from extensive surgical interventions may represent further factors that ignite depression [76,77,78]. Patients who have pre-existing psychiatric disorders should be closely monitored after a cancer diagnosis because they demonstrate a higher rate of depression relapse [79].

The prevalence of major depression in patients with advanced cancers is 5–20% [80]. Studies have also reported also a high prevalence of depression among caregivers of oncological patients (12.5–27.9%) [81]. Survival after a cancer diagnosis is lower in subjects with psychiatric comorbidities: mortality risk is 20% higher in depressed patients [82]. Both psychotherapy and pharmacotherapy have been proven useful for depression in cancer patients. While psychotherapy is effective for minor depression, pharmacotherapy is a requirement for severe depression. Shorter survival periods and the necessity for rapid effects should be carefully considered in patients with advanced disease. The preferred antidepressant drug therapies are serotonin noradrenaline reuptake inhibitors (SNRIs) and selective serotonin reuptake inhibitors (SSRIs) [83,84,85].

Neurological side effects during ACC treatment (not only mitotane) reduce compliance with treatment. Patients with normal/low mitotane plasma levels (<10–14 mg/L) can experience several neurological side effects [20], probably due to enzymatic variability and differing metabolites [16,17]. Mitotane-induced adverse effects in the central nervous system are common (although reported in only 1/10 to 1/100 patients in the ESE/ENSAT guidelines [5], they are more prevalent in clinical practice) and are characterized by loss of concentration and confusion, speech disturbance, ataxia, neuromuscular manifestations, somnolence, diminished thinking speed, depression, decreased memory, muscle tremors, polyneuropathy, vertigo and dizziness. Neurological side effects are common in clinical practice; however, evidence for them in the literature is limited. Some studies have reported the onset of dizziness, fatigue, confusion, movement and coordination disorders, memory loss, concentration difficulties and difficulty talking after short-term use of low-dose [86] or high-dose regimens [87]. In rare cases, symptoms can evolve to severe metabolic encephalopathy [88]. After the onset of mitotane-induced neurological side-effects, close monitoring of plasma levels and dose reduction are suggested. If mitotane is used during EDP or other antineoplastic treatments, the neurotoxicity is increased. Chemotherapy-induced peripheral neuropathy is reported in 50% of patients during cisplatin treatment, and recent preclinical studies have shown that the intake of cisplatin via organic cation/carnitine-mediated transporters into dorsal root ganglia neurons might trigger fasciculations, prolonged muscular contractions, paresthesia and dysesthesia [89].

### 4.5. Pain

Pain is a highly prevalent disturbing and disabling symptom in most patients with cancer. More than half of patients complain of pain following diagnosis, and its prevalence grows to up to 80% in the advanced-phase and end-of-life periods [90,91]. Cancer pain is one of the main determinants of QoL, daily activity limitations and performance status.

According to expert consensus, the prevalence of pain in neoplastic patients is high, and its intensity is mostly moderate to severe. In most patients with ACC, pain is not the first or the dominant symptom; nonetheless, it can be relevant in advanced disease. ACC pain may originate from the enlarging adrenal tumor in the retroperitoneal space directly, peritoneal carcinomatosis or bone/organ metastasis. ACC patients may be affected more often than others by painful osteoporotic bone fractures (e.g., hypercortisolism with glucocorticoid-induced osteoporosis or a skeletal disease that may be secondary to mitotane treatment [92]) and neuropathies secondary to mitotane treatment and chemotherapy.

Cancer pain treatment is based on the skillful use of non-steroidal anti-inflammatory drugs, opioids and adjuvants as neuropathic pain analgesics [51,93,94]. Wide steroid prescriptions for pain that could be reasonable for bone and visceral cancer pain should be carefully considered in ACC due to the risks of interfering with endocrine and hormonal therapy assets. Adrenal insufficiency in patients on long-term opioids must be considered [95]. Despite effective pain management with standard pharmacological therapy in over 80% of cancer patients, the prevalence of uncontrolled oncological pain is still high, reflecting current undertreatment and erroneous drug administration [96].

## 5. Simultaneous Care and Palliative Care

Palliative care is an “*approach that improves the QoL of patients and their families, address the problems associated with life-threatening illness, through the prevention and relief of suffering by means of early identification and smart assessment and treatment of pain and other problems, physical, psychosocial and spiritual*” [97,98]. In recent decades, a few disruptive studies on early management in specialized palliative care programs for advanced cancer patients have shown improved control of psychological and physical symptoms, better communication and planning of the cure and improved QoL determinants. The burden of aggressive oncological treatments and medical costs, particularly in the end-of-life period, decreased, and survival did not change (or even improve). This evidence introduced a new paradigm for palliative care, which passed from an accompaniment attitude for dying patients to a specialist and well-defined discipline aimed at symptom management, spiritual and psychosocial care, caregiver support, empathic communication and end-of-life care [99]. Therefore, evolving from the “cure” to the “care” concept, palliative care has become a form of preventative care [100,101,102]. Nowadays, opinions on the fundamental role of palliative care in modern, patient-centered, integrated cancer diagnostic and therapeutic clinical pathways have achieved a robust consensus both among experts and professional organizations. The WHO states that palliative care is applicable early in the course of illness in conjunction with other therapies that are intended as curative or life-prolonging treatments, such as chemotherapy or radiation therapy, and that it includes those investigations needed to better understand and manage distressing clinical complications [97].

Patients affected by ACC suffer from several physical symptoms and psychosocial and existential issues, and their QoL is dramatically damaged (as summarized in Figure 2). Many of these complaints are similar to those of other neoplastic patients; nonetheless, ACC patients face endocrine complications and the peculiar side effects of specific treatments. The most commonly reported mitotane-induced side effects are nausea, fatigue, loss of appetite and neurological symptoms, which affect the final compliance of the patient. In the case of severe hypercortisolism, cancer-related depression can be enhanced.

Despite the improvement in survival obtained through comprehensive, multidisciplinary treatment focused on disease [103], some disorders are still unaddressed. Palliative care treats patients as whole biopsychosocial unities, and its integration into the ACC multidisciplinary plan can complete the care pathway both during the antineoplastic treatment period (simultaneous care) and when oncological therapies are no longer effective, associated with unbearable side effects or refused by the patients. Italy’s laws guarantee the availability of palliative care for cancer patients.

There is much evidence in the literature in favor of the rapid integration of palliative care into the therapeutic path of patients with advanced cancer early on in active oncological treatments [104]. This approach, which involves the multidisciplinary collaboration of several specialists in the treatment of this category of patients (oncologist, palliative care specialist, endocrinologist, nutritionist, nurses, psychologist), is known as “simultaneous care”. This model seems to improve not only control over symptoms associated with cancer but also QoL, the costs of care, the satisfaction of patients/caregivers and survival in some cases [105,106,107,108].

The Italian Association of Medical Oncology, in accordance with the European Society of Medical Oncology and the American Society of Clinical Oncology, recommends early integration of palliative care in cancer treatments. Therefore, the correct questions are “when” palliative care must intervene in the cancer care trajectory and what the optimal model of care is [109,110], as reported in the Table 1. Instruments exist, such as the NeCPal ICO Tool [111], that can be used to detect patients who require palliative care. NeCPal combines the surprise question “Would I be surprised if this patient dies within the next (6, 12, or 24) months?” with additional clinical parameters (such as nutritional status, symptoms, limitations on/dependency of activity of daily living) and is used to identify patients with limited life prognoses [111,112]. Health-care systems are regulated by policies based on evidence-based practice: the tool helps the physician in their decision about when to switch a patient to a palliative-centered plan of care [112]. In a recent systematic review and meta-analysis, the surprise question was found to be one of the most significant predictors of mortality (HR 7.57; 95% CI 4.41–12.99) [113].

A recent study evaluated the impact of simultaneous care in a sample of 753 patients treated at a dedicated outpatient clinic (the Simultaneous Care Outpatient Clinic (SCOC)) in our high-volume Comprehensive Cancer Center [114], demonstrating the importance of close collaboration between oncologists and the palliative care team and of guarantying early access to palliative care, even during active oncological treatments. The same model confirmed the need to use indicators for the assessment of the SCOC team’s performance and to improve its organization. A further analysis evaluated the demographic characteristics, tumor site, treatment setting, survival and symptom burden using a validated reported outcome measures, identifying the categories of patients with advanced cancer, regardless of tumor types, who require special attention and quick access to simultaneous care [115].

Palliative care organizations deliver appropriate assistance and mobility aids for functionally impaired patients and psychological and spiritual support for patients and their families, both at the patient’s home (home care organizations) and residentially (hospices).

However, modern palliative medicine activities extend further than care and assistance for symptomatic patients. Palliative care time is employed proportionately to build trust with patients; understand their needs, desires, aspirations and wills; include them in the planning for optimal symptom control; and support the difficult choices that disseminate through the course of their disease. Palliative care integrates patient wishes and employs treatment options balancing QoL and increased survival while considering therapy-associated risks and complications.

Palliative care skills encompass prognostication and appropriate communication of bad news. The disclosure of a poor prognosis allows patients to better cope with their disease; if the information is given empathically, a climate of trust can be created in which patients can clarify their new priorities, reset goals and plan novel projects [116]. Knowledge of prognosis is also associated with better time management by family members in assisting their relatives in the end-of-life phases [117].

Shared decision making is a key element of cancer care that requires that the patient has sufficient knowledge of their disease and treatment options and which is greatly facilitated through the plain and empathic support of palliative medicine professionals [9]. Patients must be encouraged and supported during cancer-directed therapies to make informed decisions about their future treatments congruent with their wishes and expectations.

Advance care planning (ACP) is a fundamental process in palliative medicine in which cancer patients can build on their personal values and goals to make informed decisions for their future care [118]. ACP discussions are constantly evolving and represent a balance between patient autonomy and guidance from the healthcare team. These conversations may evolve over time and represent a balance between patient autonomy and the input and guidance from caregivers and healthcare teams [119,120].

Lastly, an important aspect of the current global health landscape is the management of health-related costs. Studies show that supporting palliative care is cost-effective because early palliative care results in minimizing expensive investigations, interventions and hospitalizations at the end-of-life period [9,97,121,122,123].

## 6. Patient-Centered Program and Palliative Care for Patients with Advanced ACC: To-Do List for 2030

According to the recommendations of scientific organizations, palliative care must be introduced early, ideally from when a patient is diagnosed—especially for patients with metastatic and incurable diseases—and continue throughout the therapeutic plan (as shown in Figure 3). In this review, we aimed to describe all the factors that should be considered when combining curative and palliative care: data regarding the positive impact of such a holistic approach are lacking, and not only for patients with ACC.

Due to the shortage of palliative medicine specialists, ACC patients with few palliative needs could be appropriately managed by their endocrinologists, oncologists and/or general practitioners (primary or basic palliative care delivery) during hospitalization, at scheduled outpatient visits and at home. Knowledge of palliative care principles and basic skills and their incorporation into clinical practice must be consolidated in the coming years (seminars, formal lessons and rotations for medical students and residents and brief visiting periods to specialized palliative care units for graduates). In the authors’ opinion, training in palliative care should be provided for all physicians/nurses involved in the natural history of patients with ACC.

Specialized (secondary) palliative care supervision and coordination of primary palliative care activities, with prompt availability for referral of difficult cases, should be guaranteed. Palliative specialists (physicians, nurses, psychologists and social workers) could participate in multidisciplinary discussions about difficult cases and provide consultations for both hospitalized patients and outpatient palliative care clinics.

The ESE and ENSAT clinical practice guidelines recommend that “*all patients with suspected and proven ACC are discussed in a multidisciplinary expert team meeting (including health care providers experienced in care of adrenal tumors, including at least the following disciplines: endocrinology, oncology, pathology, radiology, surgery) at least at the time of initial diagnosis. In addition, this team should have access to adrenal-specific expertise in interventional radiology, radiation therapy, nuclear medicine and genetics as well as to palliative care teams*” (R.1.1) and further state that “*We recommend integrating palliative care into standard oncology care for all patients with advanced ACC*” (R.10.4) [5]. Very advanced and complex ACC patients should be referred to tertiary centers early. We recently reported that early multidisciplinary evaluations can increase overall survival in patients with advanced ACC [6]. Nonetheless, curative treatments are the first part of the patient’s life: we only systematically included a simultaneous care approach and palliative care team in the multidisciplinary evaluation of ACC in recent months. Three stakeholders are currently involved in the multidisciplinary adrenal team at Padova: the University of Padova, the Padova University-Hospital and the Veneto Institute of Oncology (the latter includes the only Pain Therapy and Palliative Care with Hospice Unit in the Veneto region). Palliative specialists should be included in all multidisciplinary teams for ACC, and early and side-by-side collaboration is of the utmost importance to improve the QoL of patients in outpatient clinics, at their homes and, when out-of-treatment, in residential hospices. Fundamental to optimal ACC care is a willingness to share responsibility and decisions regarding the treatment options. Shared planning of care during multidisciplinary discussions for more complicated patients and the periodic meeting of the steering committee represent the standard of care required to continuously improve ACC care strategies [1,9,104]. Palliative and curative interventions must harmonize around unanimous objectives. The transition from primary to secondary palliative care is triggered by the complexity of both care and the prognosis. Recently developed instruments, such as the NeCPal ICO Tool [111], could aid oncologists and endocrinologists (after primary palliative care training) in detecting patients who require direct specialist palliative care [51,124].

## 7. Conclusions

To conclude, the burden of disease is often greater in patients with ACC. The standard life-prolonging treatments (surgery, mitotane and chemo- or radiotherapy) are not free of adverse events and can negatively impact the QoL of patients. Moreover, in everyday life, patients and their caregivers face several symptoms, such as nausea, pain, fatigue, anxiety, denial, bad prognoses and so on. European ESE/ENSAT [5] and AACE [3] guidelines recommend the early integration of palliative/simultaneous care in the multidisciplinary evaluation of ACC patients. Parallel treatment of both cancer and the patient is the cornerstone of simultaneous care: the secondary palliative care team should be integrated in the multidisciplinary evaluation of patients with ACC, and the other physicians should have training in primary palliative care.

## Figures and Tables

**Figure 1 cancers-14-05901-f001:**
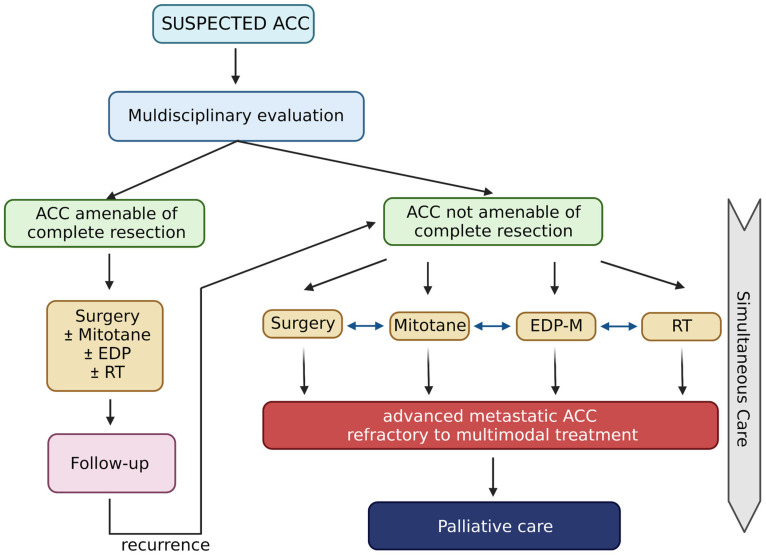
Life-prolonging curative treatment for patients with ACC. Created with BioRender.com.

**Figure 2 cancers-14-05901-f002:**
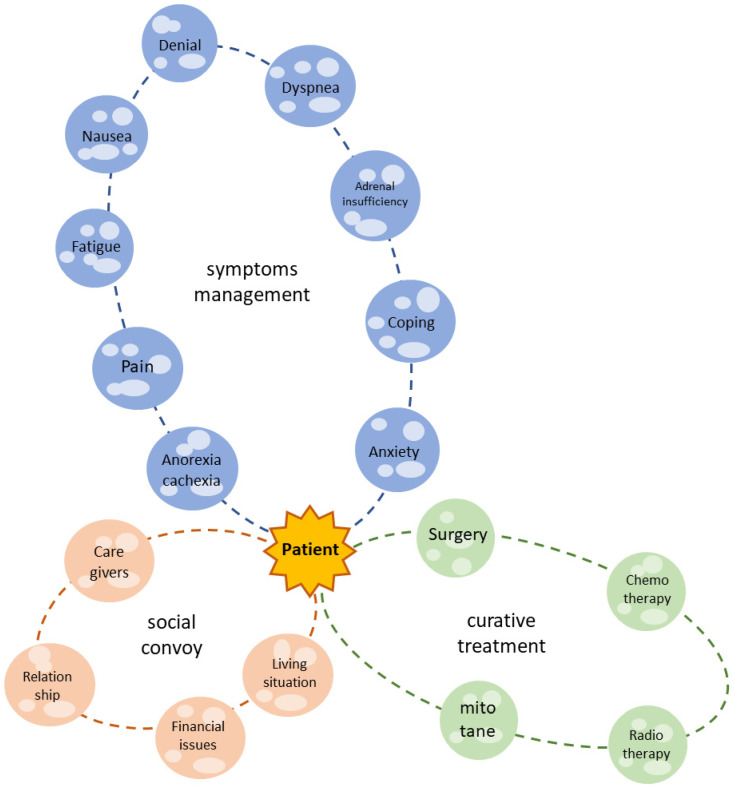
The patient with ACC is the center of the galaxy. The green solar system is what we actually consider as life-prolonging or curative treatment. Nonetheless, the patient experiences several symptoms (blue solar system) and lives in a social context (orange solar system), which must be considered in a modern, holistic, multidisciplinary approach.

**Figure 3 cancers-14-05901-f003:**
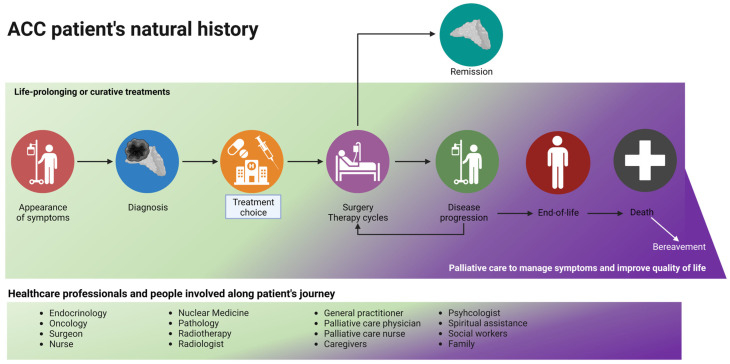
The journey of a patient with ACC: a palliative care evaluation should be considered early in the natural history of the disease, especially for patients with metastatic and advanced disease. During disease progression and the planning of second/third-line treatments, the time spent on life-prolonging treatment and on palliative care (in purple) should be equal. Curative treatment (in green) stops at the end-of-life phase, and palliative care continues after death (with family support). Created with BioRender.com.

**Table 1 cancers-14-05901-t001:** When and why to consider primary or secondary palliative care for patients with advanced ACC.

Palliative Care in ACC Patients: When?
**Both these criteria** Patients with advanced disease (stage IV) and/or requiring chemotherapy Indications from the NeCPal ICO Tool and surprise question
**Further criteria for secondary palliative care (at least one)** Limited performance status (ECOG> = 3; KPS <= 50) Superior vena cava syndrome Medullary compression Hepatic and/or renal insufficiency Effusions of neoplastic origin Severe physical, psychiatric, psychosocial or substance-abuse comorbidities Refractory pain Delirium, major depression, cachexia Other uncontrolled symptoms Severe distress related to cancer diagnosis and/or therapy Spiritual crisis and/or suicidal ideation, attempts or requests Difficulty communicating with the patient and/or his/her family Care-planning support needs
**Palliative Care in ACC Patients: Why?**
**Complex symptom management** ▪ Treatment of refractory symptoms (e.g., pain, depression, dyspnea, nausea), regardless of endocrine secretion control ▪ Complex treatments of pain and other bothering symptoms (e.g., opioid rotation, parenteral analgesics therapies, drug infusions) ▪ Help in dealing with complex situations of psychological, spiritual and/or existential suffering ▪ Palliative sedation for otherwise intractable symptoms
**Global management of complex patients** ▪ Support for loss of mobility and increased assistance needs (home and residential hospice care) ▪ Multiprofessional and multidisciplinary program of care case-management coordination
**Help in difficult decision-making processes and/or in defining treatment goals** ▪ Communication and awareness improvements ▪ Definition of care goals ▪ Discussion in moments of “transition” of care (e.g., futile treatments, surgical interventions that do not lead to an improvement in the quality of life) ▪ Management of conflicts relating to methods used for treatment objectives: ○ Within the family ○ Between families and a care team ○ Among different care teams ▪ Redefinition of “hope” in clinically and ethically complex situations ▪ Sharing of decision making and advanced care planning for the end-of-life stages

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
