# Peer review of "Advanced Adrenocortical Carcinoma: From Symptoms Control to Palliative Care"

_cancers, 2022, doi:10.3390/cancers14235901_

Round 1

Reviewer 1 Report

Subject of interest. Not much is implemeted in the field as I can see it. And this paper should therefore be rewritten especially the introduction. And be somewhat better in the implementation paragraph.

Introduction is in contrast to the rest of the paper not well written. Too vague and imprecise. Too much extremely …. en bloc resection related to good prognosis … ACC up to stage III characterized by a good prognosis and so on.

·       Page 3 lines 101-102 Reduced therapeutic range of Mitotane …. Cause of periodically assessment of level.

·       Page 4 figure 1 Mitotane monotherapy is an option for ACC not amenable of complete resection (when not aggressive)

·       Page 7, chapter nausea. Mitotane is known for inducing nausea and diarrhea. Should be described in this section. Nausea treated by f.e. metoclopramide and diarrhea by loperamide.

·       Page 7, chapter anorexia. Mitotane causes change in appetite of food, changes in tasting of food and subsequent less appetite. To be described.

·       Moreover, I miss for Mitotane the neuropsychological side effects, which are serious for a significant part of the patients especially with level over 14mg/L.

·       Pain is not a dominant symptom in ACC, other than suggested in many parts of the paper, they suffer from fatigue, anxiety, hormonal changes, loss of appetite

·       Page 8, nausea is not the major side effect affecting compliance. For some certainly, but many experience other side effects major/most important: neurological, amount of medication, big tablets, altered taste and appetite and fatigue combination dominate many patients.

·       ACC is rare. Specialized hospitals must see those patients. In such a hospital a palliative team should be present. That team should have contact with or be part of the ACC team. ACC specialists should indeed have training in palliative care.

·       In the recommendations of the ESE/ENSAT guideline 2018 (ref 5) recommendation R.1.1 and R.10.4. palliative care teams are recommended to be integrated. This should be cited as such, and in the last paragraph I advise to state that in the multidisciplinary teams for ACC palliative specialists should be included.

Author Response

Reviewer #1

Subject of interest. Not much is implemeted in the field as I can see it. And this paper should therefore be rewritten especially the introduction. And be somewhat better in the implementation paragraph.

[Reply to reviewer #1] We appreciate for Editors/Reviewers’ warm work earnestly, and hope that the correction will meet with approval. We tried our best to improve the manuscript and made some changes in the manuscript. These changes will not influence the content and framework of the paper. We look forward to hearing from you regarding our submission. We would be glad to respond to any further questions and comments that you may have. Once again, thank you very much for your comments and suggestions.

Introduction is in contrast to the rest of the paper not well written. Too vague and imprecise. Too much extremely …. en bloc resection related to good prognosis … ACC up to stage III characterized by a good prognosis and so on.

[Reply to reviewer #1]: we modified several part of the introduction according to your suggestion.

Page 3 lines 101-102 Reduced therapeutic range of Mitotane …. Cause of periodically assessment of level.

[Reply to reviewer #1]: we changed sentence and paragraph.

Page 4 figure 1 Mitotane monotherapy is an option for ACC not amenable of complete resection (when not aggressive)

[Reply to reviewer #1] we agree with the reviewer. Our aim was to combine the 2 flow-charts of the ENSAT guidelines, in order to present that simultaneous care should be considered early in patients with advanced or progressive disease. Nonetheless, we change the figure accordingly.

Page 7, chapter nausea. Mitotane is known for inducing nausea and diarrhea. Should be described in this section. Nausea treated by f.e. metoclopramide and diarrhea by loperamide.

[Reply to reviewer #1] thank so much for the suggestion, we added some sentences.

Page 7, chapter anorexia. Mitotane causes change in appetite of food, changes in tasting of food and subsequent less appetite. To be described.

[Reply to reviewer #1] we changed text accordingly, both in the dedicated paragraph and in the appropriate section of the manuscript.

Moreover, I miss for Mitotane the neuropsychological side effects, which are serious for a significant part of the patients especially with level over 14mg/L.

[Reply to reviewer #1] thank so much for the suggestion, we added the required data on mitotane, both in the dedicated paragraph and in the appropriate section of the manuscript.

Pain is not a dominant symptom in ACC, other than suggested in many parts of the paper, they suffer from fatigue, anxiety, hormonal changes, loss of appetite

[Reply to reviewer #1] we agree with the reviewer, nonetheless in some cases pain can be a dominant symptom (especially in advanced abdominal disease or in case of skeletal metastasis). We modified the paragraph accordingly, and moved the paragraph dedicated to pain at the end of the section dedicated to oncological symptoms. In the present version, fatigue (and adrenal insufficiency) is the first symptom.

Page 8, nausea is not the major side effect affecting compliance. For some certainly, but many experience other side effects major/most important: neurological, amount of medication, big tablets, altered taste and appetite and fatigue combination dominate many patients.

[Reply to reviewer #1] we agree with the reviewer, and we modified most of the manuscript accordingly.

ACC is rare. Specialized hospitals must see those patients. In such a hospital a palliative team should be present. That team should have contact with or be part of the ACC team. ACC specialists should indeed have training in palliative care.

[Reply to reviewer #1] we completely agree, this is the aim of the review. We stressed it in the first part of the manuscript, in order to introduce the paragraphs and conclusion.

In the recommendations of the ESE/ENSAT guideline 2018 (ref 5) recommendation R.1.1 and R.10.4. palliative care teams are recommended to be integrated. This should be cited as such, and in the last paragraph I advise to state that in the multidisciplinary teams for ACC palliative specialists should be included.

[Reply to reviewer #1] we added the recommendation accordingly

Reviewer 2 Report

This is a review article on adrenocortical carcinoma focusing specifically on symptoms control and palliative care role in its management. This topic is pertinent given the rarity and usually poor prognosis of adrenocortical carcinoma especially if metastatic; its common endocrine manifestations which can be difficult to diagnose and manage; and because of the rising importance of early involvement of palliative care specialists in the care of cancer patients. The article is well-structured, the figures are quite useful and easy to understand, and the references are accurate.

However, there are a few revisions that are needed.

-         Line 119: would also add mifepristone as an option

-         Line 130: would discuss mifepristone in this case

-         Line 143: would be useful to cite other cases that had challenging management of Cushing syndrome with ACC, such as: Silverman E, Addasi N, Azzawi M, Duarte EM, Huang D, Swanson B, Ganti AK, Reiser G, Fingeret AL, Kotwal A. Recurrent Cushing Syndrome From Metastatic Adrenocortical Carcinoma With Fumarate Hydratase Allelic Variant. AACE Clinical Case Reports. 2022 Sep 17.

-         Line 162: the authors mention “discussed extensively in the previous chapter”. What chapter are they referring to? Are they just referring to the earlier paragraphs? If so, I would change the wording.

-         Line 299: should be “second-line”

-        The language is poor with multiple grammatical errors, too many to be pointed out here. This needs extensive copyediting with an English writing service or I recommend authors to include another author with experience in this regard. Some examples include:

Line 16: should be “diagnosed late” and not “diagnosed lately”

Line 21: “significative” is not a word

Lines 321 and 327: “oressizing” is not a word that I am aware of.

Lines 237-238: “Moreover, high symptoms load obstacles to appropriate dosage chemotherapy and are related to reduced survival” is an odd phrasing of the sentence.

Author Response

Reviewer #2

This is a review article on adrenocortical carcinoma focusing specifically on symptoms control and palliative care role in its management. This topic is pertinent given the rarity and usually poor prognosis of adrenocortical carcinoma especially if metastatic; its common endocrine manifestations which can be difficult to diagnose and manage; and because of the rising importance of early involvement of palliative care specialists in the care of cancer patients. The article is well-structured, the figures are quite useful and easy to understand, and the references are accurate.

[Reply to reviewer #2] First of all, we thank you for such a careful, in-depth, and extensive review of our manuscript that has improved our paper by a huge margin. The modifications made as a result of your review comments will make the paper much clearer for the learned readers of this esteemed journal and make the information in it more clinically relevant.

However, there are a few revisions that are needed.

Line 119: would also add mifepristone as an option

[Reply to reviewer #2] we added it accordingly.

Line 130: would discuss mifepristone in this case

[Reply to reviewer #2] We agree with the reviewer; nonetheless, we discuss mifepristone after the part of the manuscript that reported steroidogenesis inhibitors (in order to maintain a scheme of treatment, more or less line 160).

Line 143: would be useful to cite other cases that had challenging management of Cushing syndrome with ACC, such as: Silverman E, Addasi N, Azzawi M, Duarte EM, Huang D, Swanson B, Ganti AK, Reiser G, Fingeret AL, Kotwal A. Recurrent Cushing Syndrome From Metastatic Adrenocortical Carcinoma With Fumarate Hydratase Allelic Variant. AACE Clinical Case Reports. 2022 Sep 17.

[Reply to reviewer #2] Thank so much for the suggestion, we added this interesting new paper.

Line 162: the authors mention “discussed extensively in the previous chapter”. What chapter are they referring to? Are they just referring to the earlier paragraphs? If so, I would change the wording.

[Reply to reviewer #2]: we changed the text accordingly.

Line 299: should be “second-line”

[Reply to reviewer #2]: we changed the text accordingly.

The language is poor with multiple grammatical errors, too many to be pointed out here. This needs extensive copyediting with an English writing service or I recommend authors to include another author with experience in this regard. Some examples include:

[Reply to reviewer #2]: a careful English revision has been performed.

Line 16: should be “diagnosed late” and not “diagnosed lately”

[Reply to reviewer #2]: we changed the text accordingly.

Line 21: “significative” is not a word

[Reply to reviewer #2]: we changed the text accordingly.

Lines 321 and 327: “oressizing” is not a word that I am aware of.

[Reply to reviewer #2]: we changed the text accordingly.

Lines 237-238: “Moreover, high symptoms load obstacles to appropriate dosage chemotherapy and are related to reduced survival” is an odd phrasing of the sentence.

[Reply to reviewer #2]: we changed the sentence.

Round 2

Reviewer 1 Report

Thank you for the revisions. The manuscript has improved. I have however still concerns.

They are given point by point

·       Delete at least 1x extremely in the first 2 sentences. Preferably delete both.

·       Reference for incidence is a review. Better to cite a study were incidence is studied.

·       The introduction has improved, but the order of the arguments/statements must be improved to my opinion.

Ø  That the ESE guideline is introduced here is OK, but then more in the end were the authors state that the palliative care should be present/integrated. Combine one and another.

Ø  And then rephrase the multidisciplinary team.

·       Page 2 line 82, Mitotane is …. Is redundant to line 88-89.

·       Page 2: is a racemic mixture of S-(-)- and R-(+)-mitotane. Is this true? Do we need this info. There is more to be said of the structure. I would delete this.

·       Page 2: half of the patients do not reach levels over 14? In our country 2/3 reach those levels, indeed some also over 20

·       Page 3 line 112, therapeutic range is not limited!. It is difficult to reach the level >14, and it takes time. Levels over 20 may be well tolerated. Levels over 14 will have side effects, impact on QoL, but that is not limited range.

·       Page 4 line 155: ‘dosages of the steroidogenesis inhibitors can be reduced‘ those inhibitors are probably reduced already. Mitotane is also a steroidogenesis inhibitor. Mitotane dose can be reduced after time maintaining the level of >14 due pharmacokinetics and fat reservoir.

·       Page 4 lines 185-186. Currently ADIUVO 2 trial is open studying adjuvant mitotane compared with mitotane and cycles of cisplatin en etoposide in patients with Ki67%. Authors state here that the Ki67 threshold might be >30%. I disagree, await the results of ADIUVO 2.

·       Page 5 line 201. Replace the word excellent, as it is not excellent.

·       Page 6 line 245. Everything for curative treatment? Not possible. It is the goal for stage I-III and rare stage IV. For metastasized patients cure is not possible, but treatment to attack the tumour will be given depending the condition and the shared decision of doctor and patient with family.

·       Page 10 line 430: it is not nervous symptoms. Neurological side effects, for example ataxia, loss of concentration, diminished speed of thinking, psychological, even psychiatric, look them up and give some of them, important in the line of your paper.

·       Page 13 line 550 onwards: You cannot conclude that you recently reported etc. And also you cannot conclude how you have organized your local palliative care. Conclude the article with what you found and find important. For example, palliative care is also important in ACC due to disease and treatment specific problems combined with bad prognosis. And palliative care should be integrated in the multidisciplinary teams, where you advocate training for all professionals in palliative care and availability of secondary palliative care professionals.

Author Response

Reviewer #1

Thank you for the revisions. The manuscript has improved. I have however still concerns.

They are given point by point

Delete at least 1x extremely in the first 2 sentences. Preferably delete both.

[Reply to reviewer 1] we have deleted both accordingly.

Reference for incidence is a review. Better to cite a study were incidence is studied.

[Reply to reviewer 1] we added two references.

The introduction has improved, but the order of the arguments/statements must be improved to my opinion.

Ø  That the ESE guideline is introduced here is OK, but then more in the end were the authors state that the palliative care should be present/integrated. Combine one and another.

[Reply to reviewer 1] we modified the introduction, with a link to the conclusion, accordingly.

Ø  And then rephrase the multidisciplinary team.

[Reply to reviewer 1] done.

Page 2 line 82, Mitotane is …. Is redundant to line 88-89.

[Reply to reviewer 1] we agree and we combined the two sentences.

Page 2: is a racemic mixture of S-(-)- and R-(+)-mitotane. Is this true? Do we need this info. There is more to be said of the structure. I would delete this.

[Reply to reviewer 1] done

Page 2: half of the patients do not reach levels over 14? In our country 2/3 reach those levels, indeed some also over 20

[Reply to reviewer 1] we agree with the reviewer’s opinion, because it was not clearly specified in the previous version of the manuscript. Our concept was that taken from the paper by Kerkhofs in JCEM 2013 describing that 10/20 patients on the high-dose regimen reached plasma concentrations (low than that compared with the low-dose regimen) during the first part of mitotane treatment, and not during all treatment. We changed the text accordingly.

Page 3 line 112, therapeutic range is not limited!. It is difficult to reach the level >14, and it takes time. Levels over 20 may be well tolerated. Levels over 14 will have side effects, impact on QoL, but that is not limited range.

[Reply to reviewer 1] we agree and changed accordingly.

Page 4 line 155: ‘dosages of the steroidogenesis inhibitors can be reduced‘ those inhibitors are probably reduced already. Mitotane is also a steroidogenesis inhibitor. Mitotane dose can be reduced after time maintaining the level of >14 due pharmacokinetics and fat reservoir.

[Reply to reviewer 1] our intention was to report that the balance of cortisol excess and insufficiency should be timely managed, in order to reduce symptoms and to avoid adrenal insufficiency. We added it in the manuscript.

Page 4 lines 185-186. Currently ADIUVO 2 trial is open studying adjuvant mitotane compared with mitotane and cycles of cisplatin en etoposide in patients with Ki67%. Authors state here that the Ki67 threshold might be >30%. I disagree, await the results of ADIUVO 2.

[Reply to reviewer 1] we take the indications from the indications in the ESE/ENSAT guidelines (captation of figure 3). Nonetheless, we nuanced the manuscript since it is a non-evidence-based data.

Page 5 line 201. Replace the word excellent, as it is not excellent.

[Reply to reviewer 1] done.

Page 6 line 245. Everything for curative treatment? Not possible. It is the goal for stage I-III and rare stage IV. For metastasized patients cure is not possible, but treatment to attack the tumour will be given depending the condition and the shared decision of doctor and patient with family.

[Reply to reviewer 1] we agree and we specified also these considerations in the subsequent sentence, in order to improve the manuscript. The term curative was used to consider life-prolonging surgery/mitotane/chemotherapy/radiotherapy in opposition to the role of palliative care, we changed also other part in the manuscript accordingly.

Page 10 line 430: it is not nervous symptoms. Neurological side effects, for example ataxia, loss of concentration, diminished speed of thinking, psychological, even psychiatric, look them up and give some of them, important in the line of your paper.

[Reply to reviewer 1] We have changed the sentence indicated in page 10, and also added further description of mitotane-induced and chemotherapy-induced neurological side effects in the dedicated paragraph (now depression and neurological side effects), according to your suggestion.

Page 13 line 550 onwards: You cannot conclude that you recently reported etc. And also you cannot conclude how you have organized your local palliative care. Conclude the article with what you found and find important. For example, palliative care is also important in ACC due to disease and treatment specific problems combined with bad prognosis. And palliative care should be integrated in the multidisciplinary teams, where you advocate training for all professionals in palliative care and availability of secondary palliative care professionals.

[Reply to reviewer 1] we agree and we changed the conclusion accordingly.

Reviewer 2 Report

The authors have adequately addressed my comments. The reviewed version of the manuscript after addressing comments from all reviewers is much improved and appropriate for publication. 

Author Response

Thank so much again.

Round 3

Reviewer 1 Report

Thank you for the significantly improved revised version of the manuscript.

I have now only 2 minor points:

1/ line 401. I think a mistake in citing. Only 1/10 to 1/100 experience neurological side effects? In my experience it is more 9/10 with higher levels.

2/ The apparently new instrument NeCPal ICO-Tool Ref from 2017) seems important or at least as aid in palliative care. The mentioning of this tool is only in Table 1 (delete 1, because there is no 2) and in the prefinal paragraph. Please explain somewhere in the text, preferably when the Table is used, if only in the legend.

Author Response

Reviewer #1

Thank you for the significantly improved revised version of the manuscript.

I have now only 2 minor points:

1/ line 401. I think a mistake in citing. Only 1/10 to 1/100 experience neurological side effects? In my experience it is more 9/10 with higher levels.

[Reply to reviewer 1] The side effects prevalence during mitotane treatment is that reported in the ESE/ENS@T guidelines (Table 7), in the 3rd and 4th rows there is depicted that “CNS: lethargy, somnolence, vertigo, ataxia Confusion, depression, dizziness, decreased memory” as common side effects, and in the caption there is reported the frequency (common ≥1/100 to <1/10) and the source of data  (EMEA website). We agree that in clinical practice the perceived prevalence can be much higher, we reported it in the manuscript.

2/ The apparently new instrument NeCPal ICO-Tool Ref from 2017) seems important or at least as aid in palliative care. The mentioning of this tool is only in Table 1 (delete 1, because there is no 2) and in the prefinal paragraph. Please explain somewhere in the text, preferably when the Table is used, if only in the legend.

[Reply to reviewer 1] We add more data regarding the tool and the “surprise question” in the manuscript, close to the table 1. The number of the table is correct according to the manuscript template available on the website of the journal.
